# Feasibility of Backscattering Coefficient Evaluation of Soft Tissue Using High-Frequency Annular Array Probe

**DOI:** 10.3390/s24227118

**Published:** 2024-11-05

**Authors:** Jungtaek Choi, Jeffrey A. Ketterling, Jonathan Mamou, Cameron Hoerig, Shinnosuke Hirata, Kenji Yoshida, Tadashi Yamaguchi

**Affiliations:** 1Graduate School of Science and Engineering, Chiba University, 1-33, Yayoicho, Inage-ku, Chiba 263-8522, Japan; 2Department of Radiology, Weill Cornell Medicine, 416 E 55th St., New York, NY 10022, USA; jek4011@med.cornell.edu (J.A.K.); jom4032@med.cornell.edu (J.M.); cah4016@med.cornell.edu (C.H.); 3Center for Frontier Medical Engineering, Chiba University, 1-33, Yayoicho, Inage-ku, Chiba 263-8522, Japan; shin@chiba-u.jp (S.H.); kenyoshi1980@chiba-u.jp (K.Y.)

**Keywords:** quantitative ultrasound, backscatter coefficient, annular array, reference phantom method

## Abstract

The objective of this work is to address the need for versatile and effective tissue characterization in abdominal ultrasound diagnosis using a simpler system. We evaluated the backscattering coefficient (BSC) of several tissue-mimicking phantoms utilizing three different ultrasonic probes: a single-element transducer, a linear array probe for clinical use, and a laboratory-made annular array probe. The single-element transducer, commonly used in developing fundamental BSC evaluation methods, served as a benchmark. The linear array probe provided a clinical comparison, while the annular array probe was tested for its potential in high-frequency and high-resolution ultrasonic observations. Our findings demonstrate that the annular array probe meets clinical demands by providing accurate BSC measurements, showcasing its capability for high-frequency and high-resolution imaging with a simpler, more versatile system.

## 1. Introduction

High-resolution and non-invasive imaging systems are desirable in clinical practice, and ultrasound-based imaging systems are now widely used. Ultrasound diagnostic equipment with probes of higher frequencies than conventional (e.g., 15–25 MHz center frequency) offers an excellent balance of spatial resolution, imaging depth, production cost, and safety [1]. These systems can visualize fine anatomical structures in clinical applications such as ophthalmology [2], dermatology [3,4], and cardiovascular diseases [5,6]. In recent years, diagnoses using B-mode imaging, as well as functional and qualitative information of biological tissues obtained from echo signals, have been performed. Techniques for the quantitative evaluation of biological tissues by indexing sound velocity, attenuation, and scattering are collectively known as quantitative ultrasound (QUS). Various methods for QUS can be divided into elastography to evaluate tissue stiffness [7,8,9,10,11,12,13,14], statistical analysis focusing on amplitude envelopes [15,16,17], and analysis focusing on frequency characteristics [18,19,20,21,22,23]. Although several QUS methods have been implemented using clinical instruments, basic investigations are still being conducted simultaneously.

In basic QUS studies, measurements are usually made using a single-element transducer because the robustness and stability of ultrasound transmission and reception conditions for measuring biological tissues are important. Single-element transducers can be produced at a variety of frequencies, and ultrasound beam conditions and have excellent resolution near the focal region. However, they have significant limitations in terms of depth of field (DOF) and lateral resolution, which deteriorate significantly outside the focal region. Therefore, ultrasound probes based on array technology have also been used for the development of QUS methods. QUS with array probes has already been applied in clinical practice, and in recent studies, high-frequency probes have begun to be used to measure biological tissues. With a large number of elements, array transducers enable electronic focusing to improve the DOF and construct a two-dimensional image without mechanically scanning the transducer. In the past, QUS using linear array transducers has been investigated through various methods, including evaluating the degree of fibrosis progression based on the Rayleigh distribution and multi-Rayleigh model for diffuse liver disease [24], assessing the degree of deviation from the amplitude envelope probability distribution of the echo signal, evaluating fat volume in the liver using the Nakagami distribution [25,26], and combining two Nakagami models with a healthy-liver structure filter method [27,28,29,30,31,32,33,34] to improve accuracy. However, high-frequency linear array transducers are difficult and expensive to manufacture owing to their small element sizes and high-density wire connections. Consequently, the number of clinical diagnostic devices equipped with such probes is limited.

Annular array probes are intermediates between single-element transducers and clinical array probes. The linear array probes used in existing diagnostic ultrasound systems consist of more than 100 small vibrating elements, and their design and production require very specific techniques and manufacturing costs. In addition, linear array probes for high-frequency ultrasound in the tens of MHz bands, as used in this study, are a new technology that has recently begun to be put to practical use, and the diagnostic equipment that can be equipped with these probes is limited to the high-performance models of each manufacturer. On the other hand, annular array probes can be designed specifically for each observation task, and ultrasound transmission, reception, and echo data accumulation can be performed by a simple system. Also, unlike commercial diagnostic equipment for clinical use, the accumulated echo data constitute a complete raw signal without any filtering and are significant in signal analysis. In other words, the ultrasound system including an annular array probe as proposed in this study has a simple structure, and its manufacturing cost is extremely low (even for prototypes that require a large development budget, the price is at least 1/10th of that of ordinary clinical devices) compared to advanced ultrasound diagnostic systems. The system can be easily introduced to clinical departments that are not familiar with ultrasound diagnosis. The ease of operation and user-friendly interface (optimized under the supervision of plastic surgeons and dermatologists) of our simple ultrasound system with a specially designed annular array probe would also be effective in expanding the range of clinical applications. They can extend the DOF with a relatively limited number of elements and require significantly fewer elements for imaging. New technologies and materials, including 1–3 composite technology, have been implemented in annular array probes [35,36,37,38,39,40], which are mainly used for imaging applications [41,42,43]. For instance, Brown and Lockwood designed a seven-channel, 45 MHz imaging system with a transmit beamformer and a receive beamformer [41]. The receiver beamformer sampling rate was only 105 MHz, which is insufficient for many microacoustic applications. Transmit beamforming, achieved using cables of different lengths, complicates the implementation. Hu et al. designed an imaging system for a six-channel, 43 MHz annular array transducer [42]. Ketterling et al. designed a five-channel, 40 MHz imaging system [43], where the DOF was significantly improved by incorporating synthetic focusing technology. The most common biomedical and clinical applications of annular array probes include ophthalmology, small-animal blood flow and fetal imaging, and photoacoustic measurements [44,45,46,47,48,49,50,51,52,53].

In our previous studies, we performed amplitude envelope statistics and frequency domain analyses of homogeneous scattering phantoms and excised rat livers using a 20 MHz high-frequency ultrasound annular array probe. The extended DOF achieved with annular arrays was found to improve the estimation accuracy of the QUS parameters compared to the fixed-focus case [54]. In addition, we compared and verified the backscattering coefficient (BSC) evaluation results for a layered phantom consisting of two types of media with different scattering characteristics. This comparison utilized a method that corrects based on the attenuation of each layer and a method that corrects based on the attenuation at the analysis location. The method using the average attenuation of each layer proved to be the most effective and adapted well to the DOF expansion using an annular array probe [55].

In this study, we compared the accuracy of BSC evaluation using an annular array probe with that of plane wave transducers, which are considered capable of stable BSC evaluation in tissues with a homogeneous distribution of weak scattering sources, such as the liver.

## 2. Materials and Methods

### 2.1. Tissue-Mimicking Phantoms

In biological tissues, the scattering intensity differs depending on the type of tissue being observed. This study fundamentally examined the annular array probe and signal analysis methods used to determine their effectiveness in coping with a variety of scattering conditions in the medium. Three different rectangular phantoms (8 cm wide × 2 cm long × 4 cm high) were prepared as references and evaluated. The solvents used for each phantom were 2 wt% agar (A1296; Sigma-Aldrich, MO, USA) and distilled water. The scatterers contained in each phantom were polyamide scattering spheres with diameters of 5 µm, 10 µm, 20 µm, and 40 µm (ORGASOL, 2002 EXD NAT, Arkema, Colombes, France), respectively, with a volume fraction of 0.5%. These phantoms mimic biological tissues with a homogeneous distribution of weak scattering sources, such as the liver and subcutaneous tissues.

For a comparative study of annular array probes with single-element transducers and linear array probes, three different rectangular phantoms (8 cm wide × 2 cm long × 4 cm high) were prepared as references and evaluated. The solvents used for each phantom were 2 wt% agar (A1296; Sigma-Aldrich, MO, USA) and distilled water. The scatterers contained in each phantom are polyamide scattering spheres with diameters of 5 µm, 10 µm, and 20 µm (ORGASOL, 2002 EXD NAT, Arkema, Colombes, France), respectively, with a volume fraction of 5%. These phantoms simulate subcutaneous tissue, which has a large volume fraction and scatters strongly due to lymphedema. Phantoms containing 40 µm scatterers with a volume fraction of 5% represent ultra-strong scattering media that do not occur in vivo and were therefore excluded from this study.

### 2.2. Annular Array Probe and Synthetic Focusing Method

Figure 1 shows the laboratory-made annular array probe with a center frequency of 20 MHz used in this study. A 25 µm-thick poly (vinylidene fluoride-tetrafluoroethylene) [P(VDF-TrFE)] film membrane serves as the acoustic layer of the array. This membrane was metalized on one side and bonded to a copper-clad polyimide film (flex) with degassed epoxy. The array has five equal-area annuli with a total aperture of 10 mm and a geometric focus of 31 mm. The spacing between the annuli was set to 100 µm. The annular array probe used in this study was designed for high frequency, high speed Doppler imaging and has been applied in several animal experiments [46,47]. The applicability to the evaluation of backscatter coefficients has been studied in terms of applicability to analytical models in BSC evaluation and accuracy compensation methods [54,55]. In this study, the accuracy of BSC analysis in a wide area to be evaluated in actual diagnosis was newly verified on real phantoms mimicking human skin tissues in order to verify the possibility of clinical application. We used a custom annular array described in references [36,43], and phantom construction, measurements, and signal analysis were performed at Chiba University. The size of each element and specifications of the annular array probe used in this study are listed in Table 1 and Table 2, respectively.

The DOF can be extended by applying an appropriate delay time to the echo signals acquired by each element of the annular array. This beamforming technique is known as synthetic focusing. The delay time $t_{n}$ is determined by dynamically changing the focal point f along the desired depth f in the sound-axis direction, expressed by the following Equation (1) [56]:(1)$$t_{n}=\frac{a_{n}^{2}\left(\frac{1}{R}-\frac{1}{f}\right)}{2c},$$
where R is the geometric focal length determined by the curvature, c is the sound velocity in the propagating medium, and $a_{n}$ is the root-mean-square of the inner and outer diameters of the n-th element, and the sound field characteristics of each element (obtained by the impulse response [57]) are considered. In addition, $t_{n}$ represents the delay time relative to one of the transmitting and receiving elements, and a round-trip delay $t^{tot}=t_{l}^{T}+t_{m}^{R}$ is required for the transmitting and receiving elements to focus at depth f. Here, $t_{l}^{T}$ is the delay when transmitted by the l-th element, and $t_{m}^{R}$ is the delay when received by the m-th element. The process is applied to all 25 element combinations used for transmission and reception.

Since the signals from the pairs used for transmission and reception are summarized over multiple focal regions, the algorithm for synthetic focusing (SF) is expressed by the following equation (Equation (2)):(2)$$S_{SF}\left(t\right)=\overset{q}{\underset{o=1}{\sum }}\left(\overset{n}{\underset{l=1}{\sum }}\overset{n}{\underset{m=1}{\sum }}e_{l,m}\left(t-t_{l}^{T}-t_{m}^{R}\right)\right)\omega_{o}\left(t\right)$$

In this study, $\omega_{o}\left(t\right)$ represents the square wave in the focal region o, and $e_{l,m}$ represents the radio frequency (RF) signal acquired by the combination of the l-th transmitter and m-th receiver. SF was applied to all sampled points. This process was also performed Ifor each transmitter and receiver pair (l, m). When not delayed, it can be treated as a single-element transducer with the same radius of curvature and aperture as an annular array, referred to as fixed focusing (FF) in this study.

As an example, a schematic of the SF in an annular array with three elements is shown in Figure 2. In this case, element 1 transmits the sound waves, while all elements receive them. Figure 2a shows the case where SF is applied to the acquired signal, while Figure 2b shows the case where FF is applied to the acquired signal. For FF (Figure 2b), the echo signals from each depth on the sound axis (indicated by circles) at the center of element 1 are directly added upon reception, forming a single focal distance determined by the radius of curvature of the annular array. In contrast, for SF (Figure 2a), an appropriate delay is added to the received RF signal based on the combination of transmitting and receiving elements and the depth on the sound axis. This phase alignment ensures that ultrasonic waves from each depth are in phase, enhancing amplitude corresponding to each depth and achieving a wide DOF. Post-processing was performed for each element transmission.

In this study, the echo signals were sampled at a frequency of 250 MHz. For the delay calculated using Equation (1), the delay interval is larger than that of a continuous analog signal. Therefore, a third-order spline interpolation was applied to the signal sampled at 250 MHz to simulate a signal with a sampling frequency of 2.5 GHz. After applying Equation (1) to a signal with a simulated sampling frequency of 2.5 GHz, the error in the delay interval was reduced by resetting the sampling frequency to 250 MHz. Here, the number of sample points increases simultaneously, but the additional points do not introduce new information beyond what was originally present.

Figure 3 shows a comparison of delay times between a simulated sampling frequency of 2.5 GHz and a normal sampling frequency of 250 MHz. The delay time for transmitting and receiving an element with the largest aperture is presented as an example of the combination of elements with the largest delay. The dashed line represents the ideal delay. The horizontal axis represents the distance from the vibrating surface, and the vertical axis represents the delay time at each distance. For a normal sampling frequency, the delay time between samples is rounded off owing to the long sampling interval, resulting in a staircase-like behavior. On the other hand, with simulated interpolation, the shorter sampling intervals result in fewer rounding errors between samples. These results demonstrate I have checked, and the numbers on the vertical axis of Figure 3 are already using the minus sign, not a hyphen. The possibility of interpolating and temporarily increasing the sampling frequency to achieve delay intervals closer to the ideal delay, rather than shifting the signal to the normal sampling frequency. However, this method assumes that waveform is not saturated when the RF echo signal is quantized. Therefore, if the waveform is saturated, compensating for the saturated portion may be necessary before applying the method [58].

### 2.3. Data Acquisition with Annular Array Probe

For a basic study, three-dimensional (3D) RF echo signals were observed using a laboratory-made ultrasonic scanner and an annular array probe. The center frequency, aperture diameter, and DOF were 20 MHz, 10 mm, and 31 mm, respectively. A pulse receiver (Model 5800; Olympus, Tokyo, Japan) was used to excite negative impulses to the element for transmission during echo data acquisition. The point spread function (PSF) near the focus of the annular array probe are listed in Table 3. After the echo signals were received, they were bandpass filtered at frequency in the range of 1–35 MHz using a receiver circuit in the pulser receiver. The echo signal was sampled to 12 bits using an oscilloscope (HDO6104; Teledyne, LeCroy, NY, USA) set at a sampling frequency of 250 MHz. The transducer was fixed to a triaxial linear rail (MTN100CC; Newport, CA, USA) and mechanically scanned in the lateral and slice directions. The phantom was fixed in degassed water at a temperature in the range of 22–24 °C. Echo signals were acquired by ultrasound irradiation of the top surface of each phantom. The above process is repeated 25 times with different transmitting and receiving elements to obtain echo data for 25 combinations of annular array probe elements. The scan pitch was 30 µm in both lateral and slice directions. Three-dimensional RF echo signals of 5000 pixels in depth × 301 in lateral × 301 in slice were acquired for all phantoms, and synthetic focusing was applied to ensure a wide DOF. All data acquisition and motor stage control procedures were performed using LabVIEW 2018 SP1 (National Instruments, Austin, TX, USA).

For comparison, the 3D RF echo signals of each phantom were observed using the same systems and procedures as those used in the basic study. The scan pitch was 90 µm in both lateral and slice directions. Three-dimensional RF echo signals of 8,192 pixels in depth × 101 in lateral × 101 in slice were acquired for all phantoms, and synthetic focusing was applied to ensure a wide DOF.

### 2.4. Data Acquisition with Single-Element Transducer

For comparison, 3D RF echo signals were observed using a laboratory-made ultrasonic scanner and a single-element concave transducer (PT25; TORAY, Tokyo, Japan). The center frequency, aperture diameter, and DOF were 25 MHz, 5.4 mm, and 10 mm, respectively. For echo data acquisition, a pulse receiver (Model 5800; Olympus, Tokyo, Japan) was used to excite negative impulses to the element for transmission. After the echo signals were received, they were bandpass filtered at frequencies in the range of 1–35 MHz using a receiver circuit in the pulser receiver. The echo signal was quantized to 12 bits using an oscilloscope (HDO6104; Teledyne, LeCroy, NY, USA) set at a sampling frequency of 250 MHz. The PSF near the focus of the single-element transducer is listed in Table 3. The transducer was fixed to a triaxial linear rail (MTN100CC; Newport, RI, USA) and mechanically scanned in the lateral and slice directions. The phantom was fixed in de-gassed water at 22–24 °C. Echo signals were acquired by ultrasound irradiation of the top surface of each phantom. The scan pitch was 30 µm in both the lateral and slice directions. Three-dimensional RF echo signals consisting of 8192 depth pixels × 301 lateral pixels × 301 slice pixels were acquired for all phantoms. All data acquisition and motor stage control were performed using LabVIEW 2018 SP1 (National Instruments, Austin, TX, USA).

### 2.5. Data Acquisition with Linear Array Probe

For comparison, two-dimensional RF echo signals were acquired using a research-platform scanner (Vantage256; Verasonics, Kirkland, WA, USA) and a linear array probe (L39-21gD; Verasonics, WA, USA). The center frequency was 31.25 MHz, and the element pitch and the number of elements were 0.055 mm and 128 channels, respectively. The focal depth in the elevation direction was approximately 6.5 mm. The PSF at the highest-resolution point of the linear array probe is listed in Table 3. Echo data at each angle were obtained by transmitting and receiving plane waves from the top of the phantom through a coupling gel at a sampling frequency of 100 MHz and steering the plane waves at 11 angles between −5 and +5°. The echo data from each angle were processed and compounded using compound plane-wave imaging (CPWI) [59] with synthetic aperture, and the resulting data were used for evaluation.

### 2.6. Evaluation of Speed of Sound and Attenuation Coefficient

The speed of sound and attenuation coefficient for each phantom required for BSC analysis were evaluated a priori using the reflection method. Three-dimensional RF echo signals were acquired using a single-element plane transducer (V313; Olympus, Tokyo, Japan) under the same conditions as described in the previous section. The center frequency was 15 MHz. An acrylic plate was placed on top of a sound absorber, and echo signals were acquired with and without the phantoms, maintaining the positional relationship between the transducer and the acrylic plate [60]. The transducer was positioned at the depth where the signal from the acrylic plate reached its maximum amplitude. The speed of sound was evaluated from the time of flight (TOF) based on the time difference between the maximum amplitude returned from the sample and the acrylic plate, using the following formula:(3)$$c_{p}=c_{0}\left(1+\frac{t_{ref}-t_{r}}{t_{b}-t_{s}}\right),$$
where $t_{s}$ is the TOF from the sample surface, $t_{b}$ is the TOF from the back, $t_{r}$ is the TOF from the acrylic plate after the sample is passed through, $t_{ref}$ is the TOF from the acrylic plate at the same position as during the measurement, and $c_{0}$ is the speed of sound in water. The thickness d of the sample was calculated as $d=c_{0}\left(t_{b}-t_{s}\right)/2$. The attenuation coefficient of the phantoms, α [dB/cm], was evaluated using the power spectra with and without the sample as follows:(4)$$\alpha \left(f\right)=\frac{8.686}{4d}\ln\frac{P_{s}\left(f\right)}{P_{ref}\left(f\right)},$$
where $P_{s}\left(f\right)$ is the power spectrum through the phantom and $P_{ref}\left(f\right)$ is the power spectrum without the sample. α represents the total attenuation at an arbitrary frequency f. 

### 2.7. BSC Analysis Using Reference Phantom Method

The BSC was evaluated using the reference phantom method [61]. The method assumes that the attenuation properties and BSC of a phantom serve as the reference medium, while the attenuation properties of the medium under analysis are known. By using a medium with known scattering characteristics as a reference signal, the BSC can be robustly evaluated. This correction accounts for the sound field of both the transmitting and receiving systems, especially when evaluating a medium with complex beam diffraction effects (such as with a linear array probe) or a medium featuring intricate scatterer structures (such as biological tissue). 

BSC was evaluated as
(5)$$BSC\left(f\right)=\frac{\bar{P\left(f\right)}}{\bar{P_{ref}\left(f\right)}}\frac{\bar{A_{ref}\left(f\right)}}{\bar{A\left(f\right)}}BSC_{ref}\left(f\right),$$
where $\bar{P\left(f\right)}$ represents the average power spectrum of the analyte in the region of interest (ROI) and $\bar{P_{ref}\left(f\right)}$ represents the power spectrum of the reference medium. The frequency response of the measured echo signal includes a component of attenuation that occurs during ultrasonic wave propagation. Therefore, we corrected for attenuation up to the analysis window using the attenuation $\bar{A\left(f\right)}$ of the evaluation medium and attenuation $\bar{A_{ref}\left(f\right)}$ of reference medium. $BSC_{ref}\left(f\right)$ is the theoretical value of the BSC of the reference medium, and the Faran model [62] was used to calculate this theoretical value of the BSC for the reference medium.

## 3. Results and Discussions of Basic Study

### 3.1. Basic Echo Characteristics of Each Phantom

The averaged amplitude envelopes in the depth direction for each scan line and B-mode images for each phantom are shown in Figure 4 and Figure 5, respectively. As the scatter-er density in each phantom was as low as 0.5%, Figure 4 shows that the echo signal amplitude from the phantom surface, set at a depth of approximately 26–27 mm from the annular array probe surface, became stronger as the scatterer diameter increased. Similarly, a decrease in the echo signal amplitude inside the phantom followed this trend. Figure 5 displays images normalized by the maximum amplitude of echoes from a phantom with a scatterer diameter of 40 µm. Phantoms with 5 µm and 40 µm scatterer diameters show noticeable differences in brightness reduction from the surface to depth. However, even for a phantom with a 40 µm scatter diameter, the echo amplitude remains sufficiently strong for observation at 38 mm, the maximum observation depth, due to the low scattering density.

In the frequency characteristics of each phantom shown in Figure 6, the total power of the echo signal increases with the scatterer diameter. However, the phantom with a scatterer diameter of 40 µm exhibits a lower apparent peak frequency compared to the other phantoms due to attenuation in the high-frequency band. Therefore, the frequency band selected for the BSC analysis was set to 15–23 MHz. This range corresponds to the frequencies within −12 dB from the maximum value of the power spectrum of the phantom containing 40 µm-diameter scatterers, ensuring a sufficient signal-to-noise ratio for analysis. 

### 3.2. BSC of Each Phantom

In the preliminary study, the BSC for each ROI was evaluated using an area extending 3 mm above and below the focal length of the annular array probe as the analysis region. The ROI size, serving as the analysis window, was set to 10 times the wavelength at the center frequency of the annular array probe in the depth direction and three times the lateral resolution of the annular array probe in the lateral direction. The average attenuation coefficients for each phantom, used for attenuation correction in the BSC evaluation, are listed in Table 4.

A phantom with scatter diameters of 5, 10, 20, and 40 µm was used as the reference phantom, with each subsequent phantom serving as the evaluation phantom. Evaluated BSCs are shown in Figure 7. The solid black line represents the theoretical BSC of the reference phantom evaluated using the Faran model. In cases where the reference and evaluation phantoms were identical, the BSC values evaluated from the measured echo signals and their frequency dependencies closely approximated the theoretical values. For scatterer diameters of 5, 10, and 20 µm in the reference phantom, the evaluated BSC value is larger than the theoretical value when the measured scatterer diameter in the evaluation phantom exceeded that of the reference phantom. In cases where the scatter diameter was smaller than that of the reference phantom, the BSC was correspondingly smaller. Throughout all scenarios, the frequency dependence closely aligned with the theoretical values. These results mirror trends observed in conventional theoretical BSC studies and those using single-element transducers, demonstrating that an accurate BSC evaluation, which mitigates the influence of ultrasonic beam transmission and reception characteristics, can be achieved even in fundamental studies using annular array probes. 

However, when using a phantom with a 40 µm scatterer diameter as a reference, the evaluated BSC values for phantoms with 10 µm and 20 µm scatterer diameters were found to be larger than the theoretical values. Several factors contribute to this discrepancy. High-frequency attenuation in the 40 µm phantom, as previously mentioned, and extremely small frequency components in the low-frequency band, where attenuation should be minimal, contribute to this discrepancy. In essence, the annular array probe is effective for imaging phantoms with a 40 µm scatterer diameter and low volume fractions, as well as for evaluating the BSC of similar phantoms (or biological tissues). However, it may encounter challenges when evaluating tissues with significantly different conditions. Nevertheless, this is not likely to pose a significant issue in clinical applications, as biological tissues with extremely large-diameter scatterers are generally not expected to be sparsely distributed on their own.

## 4. Results and Discussions of Comparative Study

### 4.1. Differences in Basic Echo Characteristics Among the Three Types of Sensors

Figure 8 shows the averaged amplitude envelopes of the echo signals for each scan line acquired by observing phantoms containing scatterers with diameters of 5, 10, and 20 µm, respectively, using the single-element transducer, annular array probe, and linear array probe. The single-element transducer showed a strong dependence of the amplitude envelope on the DOF, whereas the annular array probe showed a more gradual depth-dependent change in the amplitude envelope. Due to its design for observing very short distances, the linear array probe exhibited a different variation in the depth of the amplitude envelope compared to the other two sensor types. The large signal amplitudes observed at depths of 0–2 mm were near-field noise and did not indicate phantom characteristics. 

In the B-mode image shown in Figure 9, composed of echo signals acquired by the three types of sensors, the properties of the amplitude envelope shown in Figure 9 are remarkably reflected. With the single-element transducer, the echo signal beyond the DOF is extremely weak, lacking sufficient amplitude for imaging. The annular array probe reduces this problem and enables deep imaging. However, in phantoms with scatterers of 20 µm diameter, the effects of deep attenuation are more pronounced compared to the differences observed in other phantoms in terms of amplitude envelope characteristics. Furthermore, the linear array probe employs multiple transmissions/receptions and complex signal processing compared to the other two sensor types. Optimized for observing very short distances, the linear array probe achieves clear imaging of the entire observation region.

### 4.2. Comparison of Single-Element Transducer and Annular Array Probe

Figure 10 shows the frequency characteristics of the echo signal from each phantom observed using a single-element transducer and an annular array probe. For the single-element transducer, an analysis area extending 9–10.5 mm in depth and 9 mm laterally was centered around the phantom surface to align with the DOF. Similarly, the analysis area for the annular array probe was set to the same dimensions as for the single-element transducer, ensuring the geometric focus was centered. The horizontal axis of each figure shows the frequency band used in the analysis for each transducer/probe: 7–33 MHz for the single-element transducer and 8–27 MHz for the annular array probe. With the single-element transducer, increasing scatterer diameter correlated with heightened attenuation in the high-frequency band and a decrease in peak frequency. The annular array probe showed a comparable trend, although the dependence on scatterer diameter was less pronounced compared to the single-element transducer, especially noticeable for phantoms with 5 µm and 10 µm scatterer diameters. This implies that the annular array probe has higher sensitivity for acquiring echo signal compared to the single-element transducer.

Figure 11a shows the results of BSC evaluation for three phantoms using reference phantoms with scatterer diameters of 5, 10, and 20 µm, respectively, based on echo signals acquired by the single-element transducer. The ROI size for BSC analysis set to 10 times the wavelength at the center frequency of the annular array probe in the depth direction and 3 times the lateral resolution of each sensor in the lateral direction. The ROI was scanned across the entire analysis area. Average attenuation coefficients for each phantom used in attenuation correction for BSC evaluation are listed in Table 5. Figure 11b shows the results of the same BSC analysis using echo signals acquired by the annular array probe. Similar to the basic study, when the reference and evaluation phantoms had the same scatterer diameter, the BSC evaluation results closely matched theoretical values for both sensors. However, when the scatterer diameters differed between the reference and evaluation phantoms, the frequency dependence of the BSC deviated significantly from the theoretical value for all conditions with the single-element transducer. In contrast, for the annular array probe, the frequency dependence of the BSC results closely approximated for the combinations with 5 µm and 10 µm scatterer diameters. Naturally, in these cases, the BSC values deviate from the theoretical values. However, because the linear relationship between the BSC deviation and the difference in scatterer diameter between the evaluation phantom and the reference phantom aligns with theoretical expectations, this deviation can be applied as a diagnostic index for tissue evaluation. The robustness of BSC evaluation for both sensors is illustrated in Figure 12, which shows the deviation of BSC evaluation results from the theoretical values when using phantoms with 10 µm and 5 µm scatterer diameters as reference and evaluation, respectively. Compared to single-element transducers commonly used in basic BSC evaluation studies, annular array probes have achieved similar or better evaluation accuracy. However, significant deviation occurs when using a phantom with a 20 µm scatterer diameter as reference compared to other phantoms, potentially making stable BSC evaluation challenging, particularly in the high-frequency band.

### 4.3. Comparison of Linear Array Probe and Annular Array Probe

Figure 13 shows the frequency characteristics of the echo signal from each phantom observed using annular and linear array probes. For the linear array probe, an analysis area extending 4–9 mm in depth and 4.7 mm laterally was centered around the phantom’s surface to align with the DOF. In contrast, the analysis area for the annular array probe was set to the same size as that of the single-element transducer, ensuring the geometric focus was centered. The horizontal axis of each figure shows the frequency bands used in the analysis for each transducer/probe: 6–25 MHz for the annular array probe and 12–22 MHz for the linear array probe. The frequency characteristics of the annular array probe show a more pronounced difference with the scatterer diameter compared to the results in Figure 10b, likely due to the larger analysis area in the depth direction compared to the single-element transducer.

The linear array probe demonstrates high sensitivity across a broad frequency range owing to the effects of multiple plane wave transmission, reception, and CPWI. However, attenuation in the high-frequency band is still observed in phantoms with a scatterer diameter of 20 µm. 

For the BSC analysis, the ROI size was set to 10 times the wavelength at the center frequency of the annular array probe in the depth direction and three times the lateral resolution of each sensor in the lateral direction. The average attenuation coefficients for each phantom used in the attenuation correction for BSC evaluation are listed in Table 5. Figure 14a,b show the results of BSC evaluation from echo signals acquired by the annular and linear array probes, respectively, following the procedure outlined in the previous section. Consistent with previous studies, when the scatterer diameters of the reference and evaluation phantoms match, the evaluated BSC values closely approximate the theoretical values. In addition, when a phantom with a scatterer diameter of 20 µm serves as the reference, the frequency dependence of the evaluation results with the annular array probe tends to align closely with the theoretical values, even when the scatterer diameter of the evaluation phantom differs. An interesting observation is that the annular array probe demonstrates greater accuracy compared to linear array probes, typically considered robust for spatially uniform resolution ultrasound and BSC evaluation. However, the linear array probe excels in assessing BSC stability when using a phantom with a small scatterer diameter as the reference and evaluating a phantom with a large scatterer diameter. In cases where the reference phantom has a small scatterer diameter and the evaluation phantom has a large scatterer diameter, particularly in terms of frequency dependence, the linear array probe shows superior stability in BSC evaluation.

Figure 15 shows the deviation of BSC evaluation values from the theoretical values when using phantoms with scatterer diameters of 10 µm and 5 µm as reference and evaluation, respectively, for comparison with the study in the previous section. The robustness of BSC evaluation using echo data acquired with the annular array probe across a wide area near the focus was confirmed.

## 5. Conclusions

Conventionally, it has been assumed that annular array probes are not suitable for QUS because the ultrasound beam cannot be mechanically and electrically controlled by the transmitter/receiver system. However, this study shows that BSC can be evaluated with accuracy comparable to that of linear array probes, which are capable of advanced beamforming, within an extended DOI range. In current clinical ultrasound diagnostics, phantoms with low scattering intensity (small scatterer diameters) serve as reference standards, while tissues with higher scattering intensity, like the liver, are evaluated using linear array probes. However, as highlighted in this study, significant deviations in frequency dependence or average BSC values occur when the scattering properties of reference and evaluation media differ significantly, impacting the reliability of tissue characterization. The annular array probe faces similar challenges, but using a reference medium with high scattering intensity showcases its potential for evaluating a wide range of biological tissues. Given their ease of design and production, annular array probes are poised to contribute significantly across various fields, especially when tailored for specific diagnostic needs in conjunction with biological tissues. One of the problems for clinical application among the issues for future study is that there are multiple types of scatterers in actual biological tissues, and the Faran model used for BSC evaluation in this study may deviate from the actual scatterer structure. However, this problem is not limited to annular array probes, but occurs in any ultrasonic signal analysis using any sensor, so a comprehensive study on the matching of theory (numerical models) and measurement (actual echo analysis) should be continued. The challenges limited to annular array probes are the limited length of the DOI due to the small number of sensors and the difficulty of ensuring observation accuracy in deep areas. It is assumed that these problems can be addressed by varying the maximum diameter of the sensor and the thickness of each ring depending on the depth and size of the area to be observed. We are currently designing a five-channel annular array probe with a maximum diameter of 30 mm for verification and will promote further studies in phantoms and real skin tissues in the future.

## Figures and Tables

**Figure 1 sensors-24-07118-f001:**
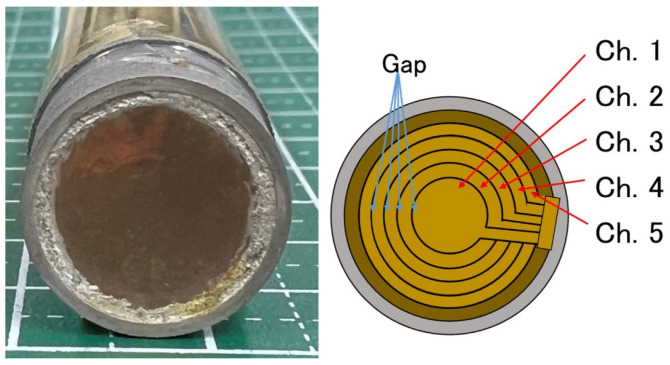
Appearance and configuration of the annular array probe.

**Figure 2 sensors-24-07118-f002:**
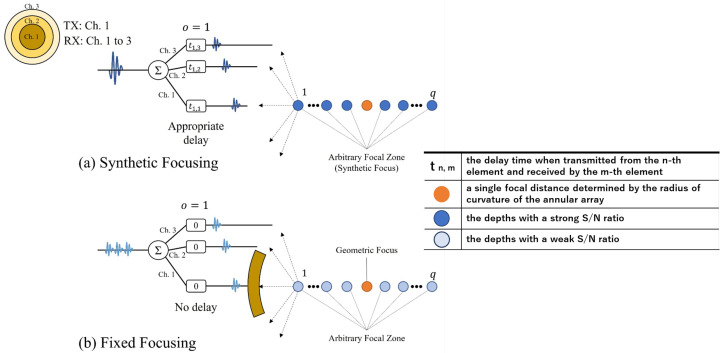
Schematic of synthetic focusing.

**Figure 3 sensors-24-07118-f003:**
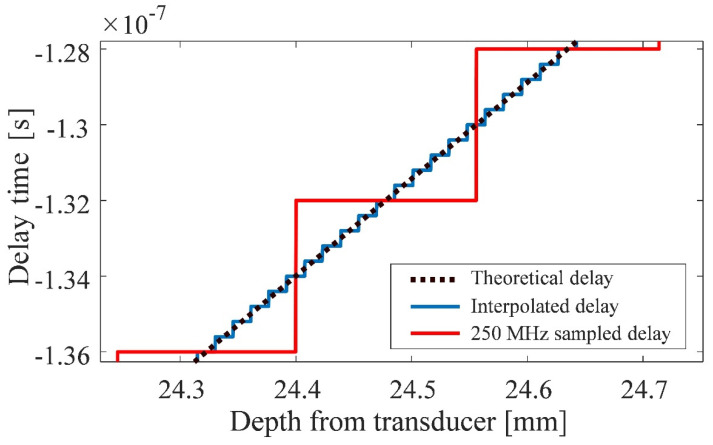
Difference in delay time due to interpolation.

**Figure 4 sensors-24-07118-f004:**
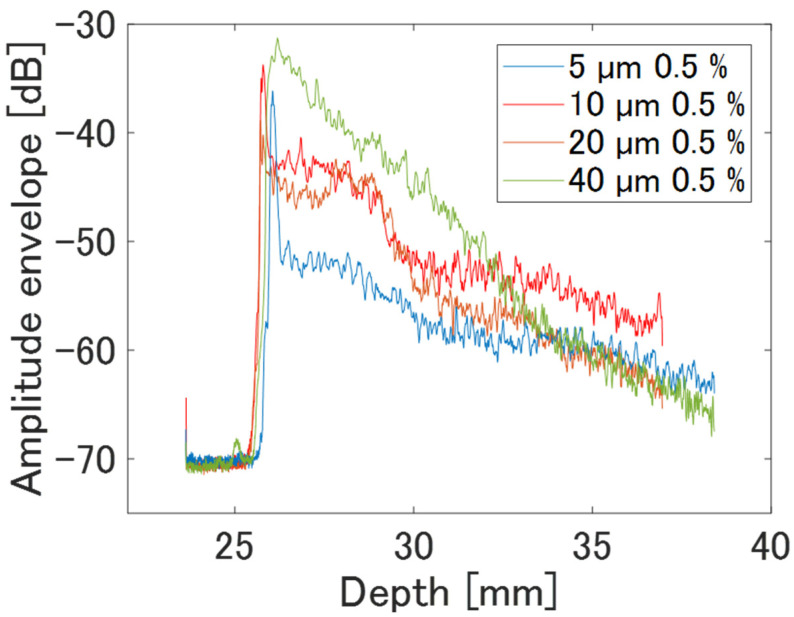
Averaged amplitude envelopes of phantoms of basic study.

**Figure 5 sensors-24-07118-f005:**
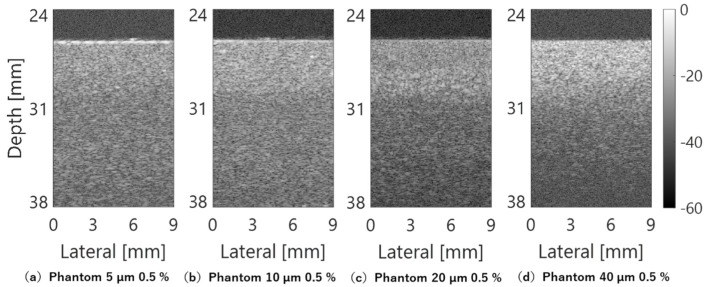
B–mode images of phantoms.

**Figure 6 sensors-24-07118-f006:**
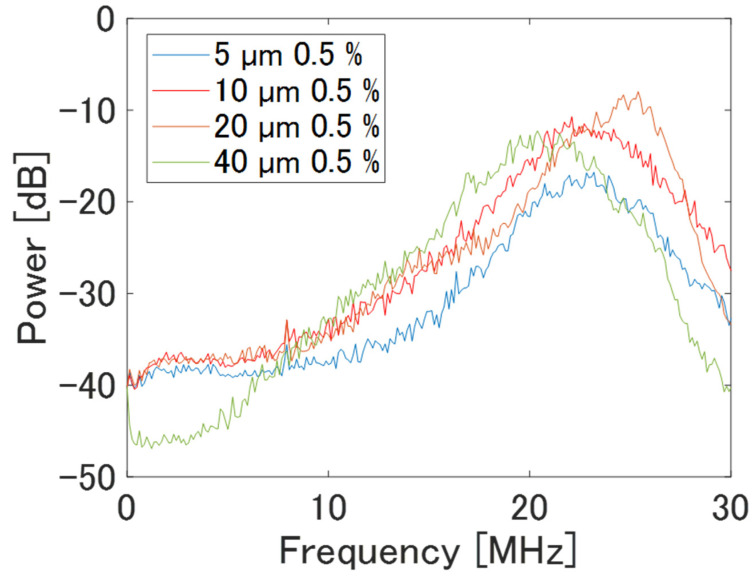
Frequency characteristics of phantoms of basic study.

**Figure 7 sensors-24-07118-f007:**
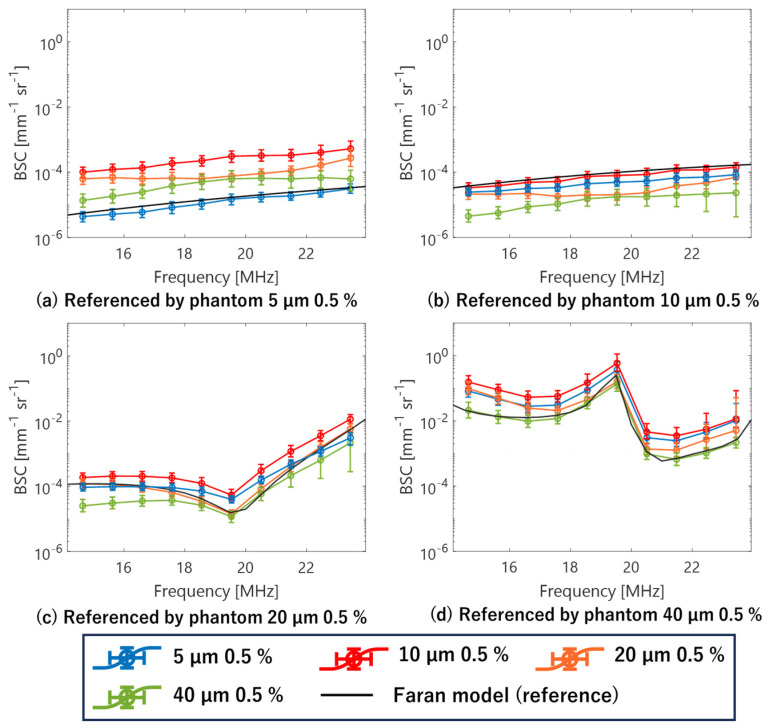
Evaluated BSCs of phantoms.

**Figure 8 sensors-24-07118-f008:**
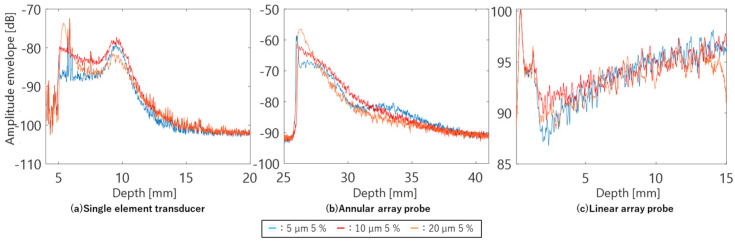
Averaged amplitude envelopes of phantoms of comparative study.

**Figure 9 sensors-24-07118-f009:**
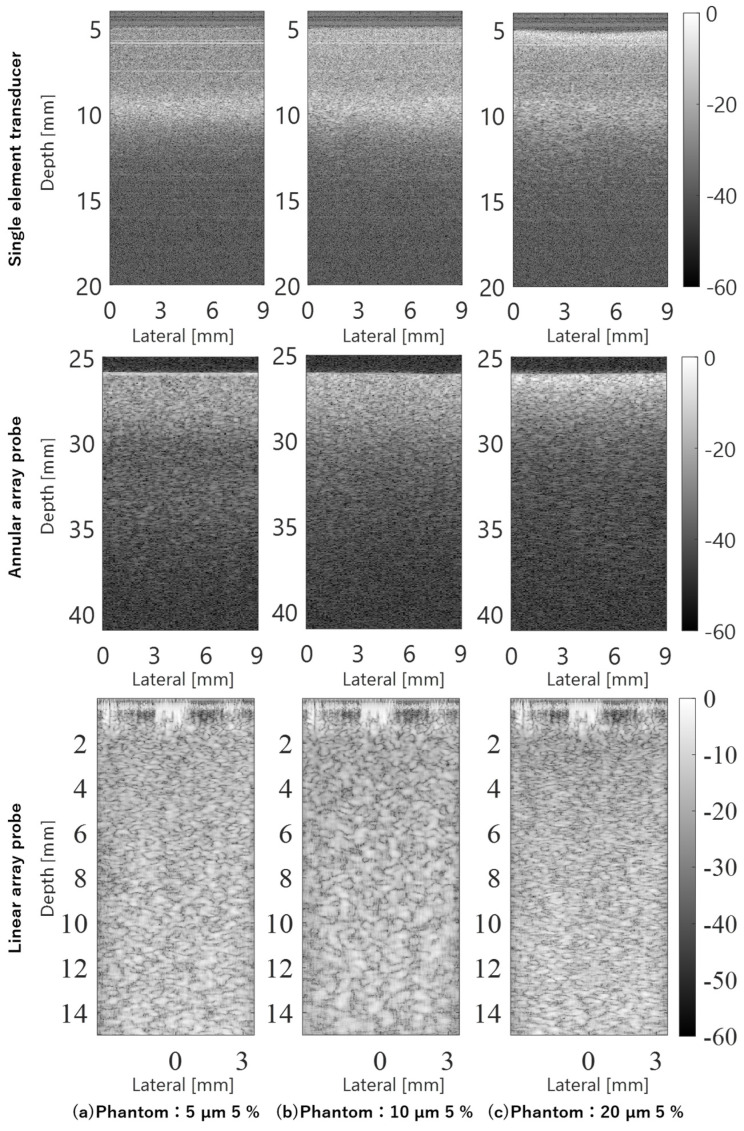
B–mode images of phantoms acquired using different ultrasound probes.

**Figure 10 sensors-24-07118-f010:**
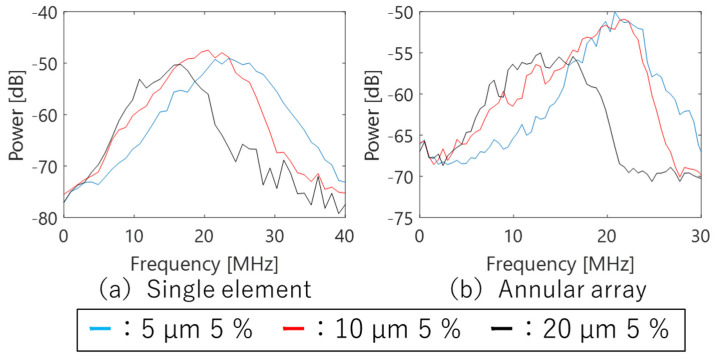
Frequency characteristics of phantoms observed using different ultrasound probes.

**Figure 11 sensors-24-07118-f011:**
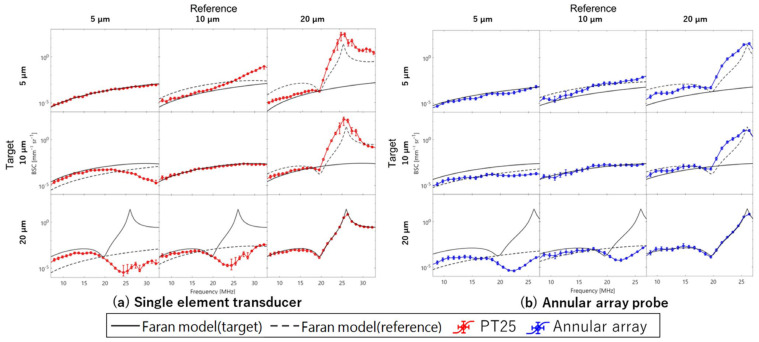
Evaluated BSCs of phantoms ((**a**) single element transducer; (**b**) annular array probe).

**Figure 12 sensors-24-07118-f012:**
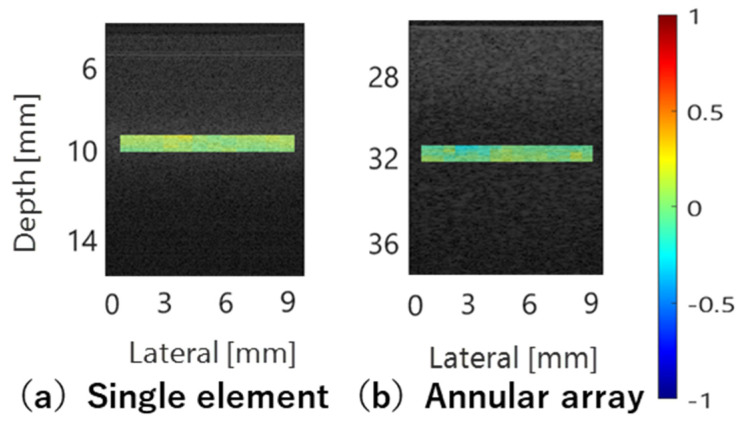
Deviation of evaluated BSCs from theoretical values in comparative study within single element transducer and annular array probe.

**Figure 13 sensors-24-07118-f013:**
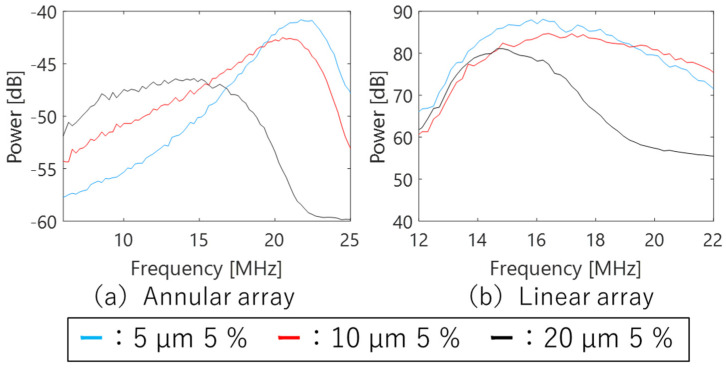
Frequency characteristics of phantoms of comparative study.

**Figure 14 sensors-24-07118-f014:**
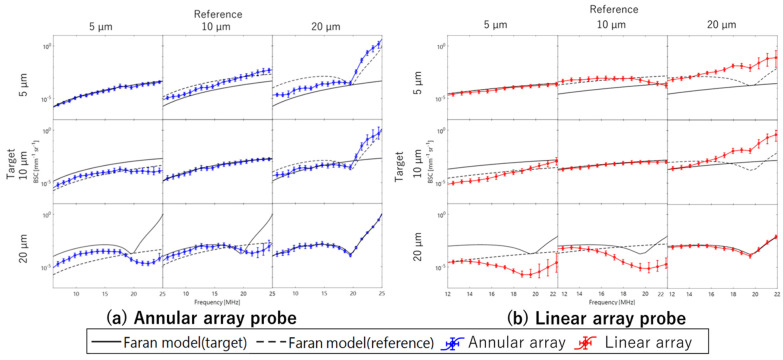
Evaluated BSCs of phantoms ((**a**) annular array probe; (**b**) linear array probe).

**Figure 15 sensors-24-07118-f015:**
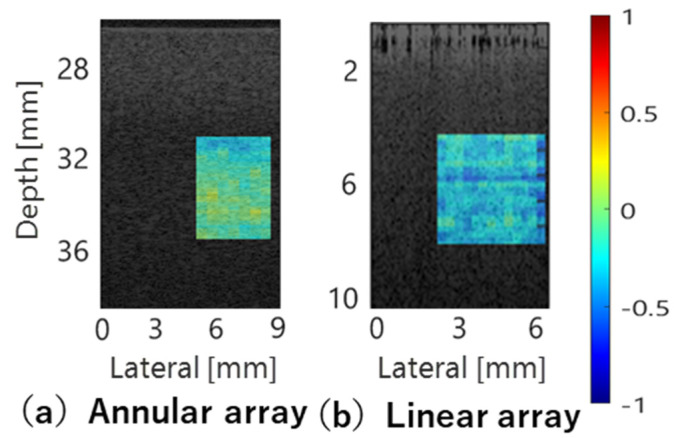
Deviation of evaluated BSCs from theoretical values in comparative study within annular array probe and linear array probe.

**Table 1 sensors-24-07118-t001:** Dimensions of annular array elements.

	Inner Radius (µm)	Outer Radius (µm)	Area (mm^2^)
Ch.1	―	2.12	14.07
Ch.2	2.22	3.06	14.03
Ch.3	3.16	3.80	14.00
Ch.4	3.90	4.44	13.95
Ch.5	4.53	5.00	13.92

**Table 2 sensors-24-07118-t002:** Technical specifications of the annular array.

**Center frequency (MHz)**	20
**Material**	P(VdF-TrFE)
**Aperture (mm)**	10
**Radius of curvature (mm)**	31

**Table 3 sensors-24-07118-t003:** PSF near focus for each sensor.

	Axial (µm)	Lateral (µm)
Annular array	75	180
PT25	53	104
L39-21gD	80	120

**Table 4 sensors-24-07118-t004:** Average attenuation coefficients for each phantom of basic study.

	Phantom (a)	Phantom (b)	Phantom (c)	Phantom (d)
Diameter of scatters (µm)	5	10	20	40
Att. coefficient (dB/cm/MHz)	0.139	0.144	0.144	1.026

**Table 5 sensors-24-07118-t005:** Average attenuation coefficients for each phantom of comparative study.

	Phantom (a)	Phantom (b)	Phantom (c)
Diameter of scatters (µm)	5	10	20
Att. coefficient (dB/cm/MHz)	0.471	0.571	1.078

## Data Availability

Data are contained within the article.

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
