# Peer review of "Feasibility of Backscattering Coefficient Evaluation of Soft Tissue Using High-Frequency Annular Array Probe"

_sensors, 2024, doi:10.3390/s24227118_

Round 1
Reviewer 1 Report
Comments and Suggestions for Authors
This study aims to evaluate the backscattering coefficient of several tissue-mimicking phantoms utilizing these different ultrasonic probes. The current manuscript could be reconsidered if the following questions are addressed.
1. Emphasizing the innovativeness and research significance of this manuscript.
2. Some related figures could be combined and rearranged into a new figure.
3. Some additional discussion is needed to compare the findings in this paper with those that have already been reported.
Comments on the Quality of English LanguageThe quality of English language is fine.
Author Response
Dear Editor and Reviewers,
Thank you for your appropriate and detailed comments on our submitted paper.
We are pleased that you are excited about the possibilities of BSC evaluation using our proposed annular array probe, and we appreciate your valuable comments that will help us to make this an even better paper.
We have responded to each of the comments as follows. We have also revised the text and figures for each of them. 
We would be grateful if you could review it again.
Best Regards,
Jungtaek CHOI
This study aims to evaluate the backscattering coefficient of several tissue-mimicking phantoms utilizing these different ultrasonic probes. The current manuscript could be reconsidered if the following questions are addressed.
Comments1. Emphasizing the innovativeness and research significance of this manuscript.
Response1.
The purpose of this study Is to investigate the possibility of clinical application of QUS techniques for shallow subcutaneous tissues using annular array probes. In this study, the possibility of evaluating backscattering coefficient with an annular array probe for scattering media assuming lymphedema was verified in comparison with a linear array probe and a single element transducer. Annular array probes have been used in many fields such as ophthalmology, small animals, and photoacoustics, and in recent years, they have been applied not only to imaging but also to QUS for ophthalmology. However, the application of array probes to QUS for soft tissues such as human skin and tumors is questionable because of the limited conditions for transmitting and receiving ultrasound beams compared to linear array probes, and at present, insufficient studies have been conducted. Naturally, the use of linear array probes is effective for QUS of these tissues in terms of penetration and resolution, but it is extremely difficult to use expensive high-performance ultrasound systems in plastic surgery, dermatology, and gastrointestinal surgery, and QUS technology with linear array probe has not been widely used. Ultrasound diagnosis using an annular array probe enables real-time data collection with a simple system configuration. If QUS with performance comparable to that of linear array probes is confirmed to be possible with annular array probes, it could provide a useful technology for many clinical departments where high-frequency ultrasound applications have been long awaited but not yet put into practical use. 
In this study, the accuracy of backscattering property analysis using an annular array probe was compared with that of a linear array probe, which has the highest accuracy in terms of spatial stability of ultrasonic measurement, and that of a single transducer, which has the highest sensitivity in a very narrow range, and the QUS The QUS adaptability of the annular array probe was verified.
The above explanation was described in Lines *516-529* in the main text.
Comments2. Some related figures could be combined and rearranged into a new figure.
Response2.
As you noted, it was possible to put together several figures, so we put Figures 11 and 12 together to create Figure 11 and Figures 15 and 16 together to create Figure 14.
Comments3. Some additional discussion is needed to compare the findings in this paper with those that have already been reported.
Response3.
In order to compare the results with those of the latest similar studies, it is necessary to compare the annular array probe with the linear array probe as in this paper, using a real-life subject such as a rat model of a disease. However, since it would take a great deal of time to discuss these additional studies in this paper, they will be additionally discussed in a future report of research results.
Reviewer 2 Report
Comments and Suggestions for Authors
This paper presents a study aimed at enhancing tissue characterization in abdominal ultrasound diagnosis through a simplified system. It evaluates the backscattering coefficient (BSC) of tissue-mimicking phantoms using three different ultrasonic probes: a single-element transducer, a linear array probe, and an annular-array probe.
This manuscript requires revision based on the following raised comments.
1. What is the main objective of this study?
2. Please list the contribution for the proposed study
3. It is suggested to include tye following research in the introduction:-
A. Elhanashi, S. Saponara and Q. Zheng, "Classification and Localization of Multi-Type Abnormalities on Chest X-Rays Images," in IEEE Access, vol. 11, pp. 83264-83277, 2023, doi: 10.1109/ACCESS.2023.3302180. keywords: {X-ray imaging;Biomedical imaging;COVID-19;Object detection;Diseases;Pulmonary diseases;Deep learning;Deep learning;multi-classification;localization;ensemble model;bronchopneumonia/lung abnormalities},
-
4. The proposed approach should be compared to the state of the art methodologies.
5. What were the key findings when comparing the annular and linear array probes?
6. What method was applied to the echo signals sampled at 250 MHz? Explain this with more details
7. The limitation and chellenge of this research should be discussed in this manuscript
8. What are the future direction for the poposed approach
Author Response
Dear Editor and Reviewers,
Thank you for your appropriate and detailed comments on our submitted paper.
We are pleased that you are excited about the possibilities of BSC evaluation using our proposed annular array probe, and we appreciate your valuable comments that will help us to make this an even better paper.
We have responded to each of the comments as follows. We have also revised the text and figures for each of them. 
We would be grateful if you could review it again.
Best Regards,
Jungtaek CHOI
This paper presents a study aimed at enhancing tissue characterization in abdominal ultrasound diagnosis through a simplified system. It evaluates the backscattering coefficient (BSC) of tissue-mimicking phantoms using three different ultrasonic probes: a single-element transducer, a linear array probe, and an annular-array probe.
This manuscript requires revision based on the following raised comments.
Comments 1. What is the main objective of this study?
Response1.
The purpose of this study Is to investigate the possibility of clinical application of QUS techniques for shallow subcutaneous tissues using annular array probes. In this study, the possibility of evaluating backscattering coefficient with an annular array probe for scattering media assuming lymphedema was verified in comparison with a linear array probe and a single element transducer. Annular array probes have been used in many fields such as ophthalmology, small animals, and photoacoustics, and in recent years, they have been applied not only to imaging but also to QUS for ophthalmology. However, the application of array probes to QUS for soft tissues such as human skin and tumors is questionable because of the limited conditions for transmitting and receiving ultrasound beams compared to linear array probes, and at present, insufficient studies have been conducted. Naturally, the use of linear array probes is effective for QUS of these tissues in terms of penetration and resolution, but it is extremely difficult to use expensive high-performance ultrasound systems in plastic surgery, dermatology, and gastrointestinal surgery, and QUS technology with linear array probe has not been widely used. Ultrasound diagnosis using an annular array probe enables real-time data collection with a simple system configuration. If QUS with performance comparable to that of linear array probes is confirmed to be possible with annular array probes, it could provide a useful technology for many clinical departments where high-frequency ultrasound applications have been long awaited but not yet put into practical use. 
In this study, the accuracy of backscattering property analysis using an annular array probe was compared with that of a linear array probe, which has the highest accuracy in terms of spatial stability of ultrasonic measurement, and that of a single transducer, which has the highest sensitivity in a very narrow range, and the QUS The QUS adaptability of the annular array probe was verified.
The above explanation was described in Lines *516-529* in the main text.
Comments 2. Please list the contribution for the proposed study
Response2.
The annular array probe used in this study was designed for high-frequency, fast-doping imaging and has been applied in several animal experiments [47,48]. The applicability to the evaluation of backscatter coefficients has been studied in terms of applicability to analytical models in BSC evaluation and accuracy compensation methods [55,56]. In this study, the accuracy of BSC analysis in a wide area to be evaluated in actual diagnosis was newly verified on real phantoms mimicking human skin tissues in order to verify the possibility of clinical application. Our probe was designed and produced at Weill Cornell Medicine, and phantom construction, measurements, and signal analysis were performed at Chiba University.
The above explanation was described in Lines *140-148* in the main text.
Comments 3. It is suggested to include the following research in the introduction:-
- Elhanashi, S. Saponara and Q. Zheng, "Classification and Localization of Multi-Type Abnormalities on Chest X-Rays Images," in IEEE Access, vol. 11, pp. 83264-83277, 2023, doi: 10.1109/ACCESS.2023.3302180.
keywords: {X-ray imaging;Biomedical imaging;COVID-19;Object detection;Diseases;Pulmonary diseases;Deep learning;Deep learning;multi-classification;localization;ensemble model;bronchopneumonia/lung abnormalities},-
Response3.
We have carefully reviewed the proposed paper and found that the content of the paper is very different from our study on ultrasound, quantitative diagnosis, backscatter coefficient, beamforming, lymphedema, etc. Therefore, it was unfortunately decided not to cite the proposed paper in the revised manuscript.
However, as you pointed out, it is necessary to add the papers related to this study in a broad sense, so the following papers are additionally cited.
[22]Jeon, S. K.; Lee, J. M.; Byun, Y.H.; Jee, J. H.; K, M.; Development and validation of multivariable quantitative ultrasound for diagnosing hepatic steatosis. Scientific Reports, Vol. 13(1), doi: 10.1038/s41598-023-42463-w
[23]Nguyuen, T. N.; Podkowa, A. S.; Park, T.H.; Miller, R. J.; Do, M. N.; Oelze, M. L.; Use of a convolutional neural network and quantitative ultrasound for diagnosis of fatty liver. Ultrasound Med Biol. 2021, Vol. 47(3), pp. 556-568
Comments 4. The proposed approach should be compared to the state of the art methodologies.
Response4.
As noted in our response to comment 3, we have cited several additional papers.
Comments 5. What were the key findings when comparing the annular and linear array probes?
Response5.
Conventionally, it has been assumed that annular array probes are not suitable for QUS because the ultrasound beam cannot be mechanically and electrically controlled by the transmitter/receiver system. However, this study shows that BSC can be evaluated with accuracy comparable to that of linear array probes, which are capable of advanced beamforming, within an extended DOI range.
The above explanation was described in Lines *497-501* in the main text.
Comments 6. What method was applied to the echo signals sampled at 250 MHz? Explain this with more details
Response6.
  The details of the echo data acquisition method using an annular array probe in this study are described below.
The target is fixed in degassed water, and ultrasonic waves are irradiated on the top surface of the target to acquire echo signals. However, the ultrasonic wave transmitter/receiver system currently in use is not capable of acquiring echo data from all 25 combinations of transmitter/receiver elements of the annular array probe in a single ultrasonic wave transmission/reception. Therefore, the 25-echo data were acquired by changing the transmitter/receiver elements for each measurement. The synthetic focusing method was then applied to the acquired echo data, which were used as evaluation data.
The above explanation was described in Lines *227-229* in the main text.
Comments 7. The limitation and challenge of this research should be discussed in this manuscript
Response7.
The problem for clinical application is that there are multiple types of scatterers in actual biological tissues, and the Faran model used for BSC evaluation in this study may deviate from the actual scatterer structure. However, this problem is not limited to annular array probes, but occurs in any ultrasonic signal analysis using any sensor, so a comprehensive study on the matching of theory (numerical models) and measurement (actual echo analysis) should be continued. The challenges limited to annular array probes are the limited length of the DOI due to the small number of sensors and the difficulty of ensuring observation accuracy in deep areas. It is assumed that these problems can be addressed by varying the maximum diameter of the sensor and the thickness of each ring depending on the depth and size of the area to be observed. We are currently designing a 5-channel annular array probe with a maximum diameter of 30 mm for verification, and will promote further studies in phantoms and real skin tissues in the future.
The above explanation has been added at the end of Conclusion.
Comments 8. What are the future direction for the proposed approach
Response8.
As mentioned in our response to comment 7, we will promote further studies using the newly designed annular array probe and mathematical models for BSC evaluation.
We will also discuss the integration of ICG with fluorescence contrast and photoacoustics, which we are considering separately, but since these are different from the purpose of this study, we will not explain them in the main text, but only in our response to the reviewers.
Reviewer 3 Report
Comments and Suggestions for Authors
1. Detailed Cost-Benefit Analysis
- The discussion on the cost-effectiveness of the annular-array probe could be expanded. By incorporating a more detailed cost-benefit analysis that compares the manufacturing costs, maintenance, and potential clinical outcomes between the annular-array probe and traditional probes, the manuscript could offer a clearer picture of the economic value of this technology in clinical settings. This would provide readers with a more comprehensive understanding of the financial implications of adopting this technology.
2. Enhanced Statistical Analysis of BSC Data
- The current analysis of the relationship between BSC deviations and scatterer diameters can be further refined. Applying more sophisticated statistical models or machine learning techniques could improve the predictive accuracy of the BSC evaluations and enhance the robustness of the study’s findings. This would allow the manuscript to offer deeper insights into the data and present more reliable conclusions.
3. Discussion of Potential Methodological Limitations
- A more detailed discussion of the potential limitations within the experimental setup or data interpretation would greatly benefit the manuscript. Addressing issues such as assumptions made during the BSC calculations, the consistency of phantom materials, or the impact of specific probe characteristics on the results would provide a more balanced view of the study’s reliability. This critical evaluation would help readers understand the context and scope of the findings better.
4. Addressing High-Frequency Attenuation Issue
- The manuscript could be strengthened by discussing possible alternative approaches to mitigate high-frequency attenuation challenges, particularly in phantoms with larger scatterer diameters. Exploring different probe designs, materials, or advanced signal processing techniques could provide practical solutions to these challenges, making the study’s conclusions more robust and applicable to various scenarios.
5. Exploration of Alternative Probe Configurations
- Consider including a discussion on the potential of alternative configurations for the annular-array probe. The manuscript could explore the theoretical impact of varying the number of elements in the array, different element arrangements, or different materials used in the probe’s construction. Such an exploration could broaden the scope of the study, suggesting new directions for future research and development.
6. Development of a More User-Friendly Interface for Probe Operation
- The manuscript would benefit from discussing the development of a more intuitive and user-friendly interface for the annular-array probe. This could involve integrating real-time data processing capabilities, providing immediate feedback to the operator, or simplifying the interface to enhance usability in clinical settings. Addressing this aspect could increase the practical relevance of the study and encourage broader adoption of the technology.
These suggestions aim to enhance the manuscript by deepening the analysis, addressing potential limitations, and proposing practical improvements. Incorporating these elements would provide a more comprehensive and impactful contribution to the field.
In the following, it is not mandatory.
If you are considering the possibility of conducting additional experiments to further enhance the study, it would be valuable to explore a few key areas. Below are suggestions for potential experiments that could provide deeper insights and strengthen the findings:
1. Broader Range of Tissue-Mimicking Phantoms
• Objective: To validate the generalizability of the annular-array probe across a wider variety of tissue types.
• Proposed Experiment: Create and evaluate additional phantoms that mimic a broader spectrum of biological tissues, including more complex and heterogeneous structures. This could include phantoms with varying degrees of scatterer density, size, and distribution to better simulate real tissue conditions and assess the probe’s performance across different scenarios.
2. Longitudinal Stability of BSC Measurements
• Objective: To evaluate the stability and repeatability of BSC measurements over time.
• Proposed Experiment: Conduct a longitudinal study where the BSC of a specific phantom is measured at multiple time points. This would help determine the consistency and reliability of the BSC values over time, which is crucial for monitoring disease progression or the effects of treatments.
3. Comparison with In Vivo Measurements
• Objective: To bridge the gap between in vitro phantom studies and real-world clinical applications.
• Proposed Experiment: If feasible, perform preliminary in vivo studies using animal models or human subjects to compare the BSC measurements obtained from phantoms with those obtained from actual biological tissues. This would provide a clearer understanding of how well the annular-array probe performs in a clinical setting and whether the results from phantom studies can be reliably extrapolated to human tissues.
4. High-Frequency Attenuation Mitigation
• Objective: To explore solutions for the high-frequency attenuation issues observed in certain phantoms.
• Proposed Experiment: Experiment with different probe materials, configurations, or signal processing techniques to see if these can reduce attenuation effects at higher frequencies. This could involve testing alternative membrane materials or adjusting the geometry of the annular-array probe to optimize performance in challenging conditions.
5. Alternative Probe Configurations
• Objective: To determine the optimal configuration of the annular-array probe for various clinical applications.
• Proposed Experiment: Test different configurations of the annular-array probe, such as varying the number of elements, the arrangement of those elements, or the curvature of the array. By systematically varying these parameters, it would be possible to identify configurations that offer the best balance of resolution, depth of field, and sensitivity for specific types of tissue.
6. Evaluation of User Interface Usability
• Objective: To assess and improve the usability of the probe’s interface in clinical settings.
• Proposed Experiment: Conduct a usability study with clinicians to evaluate the current interface of the annular-array probe. Collect feedback on ease of use, real-time data interpretation, and workflow integration. Based on this feedback, develop and test improvements that could enhance the user experience and potentially increase the adoption of the technology in clinical practice.
If any of these proposed experiments are feasible, they could significantly enhance the robustness of the study and provide more comprehensive data to support the conclusions. Please let me know if you would like to explore these options further or if there are specific aspects you are interested in investigating.
Author Response
Dear Editor and Reviewers,
Thank you for your appropriate and detailed comments on our submitted paper.
We are pleased that you are excited about the possibilities of BSC evaluation using our proposed annular array probe, and we appreciate your valuable comments that will help us to make this an even better paper.
We have responded to each of the comments as follows. We have also revised the text and figures for each of them. 
We would be grateful if you could review it again.
Best Regards,
Jungtaek CHOI
Comments 1. Detailed Cost-Benefit Analysis
- The discussion on the cost-effectiveness of the annular-array probe could be expanded. By incorporating a more detailed cost-benefit analysis that compares the manufacturing g costs, maintenance, and potential clinical outcomes between the annular-array probe and traditional probes, the manuscript could offer a clearer picture of the economic value of this technology in clinical settings. This would provide readers with a more comprehensive understanding of the financial implications of adopting this technology.
Response1.
The linear array probes used in existing diagnostic ultrasound systems consist of more than 100 microscopic vibrating elements, and their design and production require very specific techniques and manufacturing costs. In addition, linear array probes for high-frequency ultrasound in the tens of MHz band, as used in this study, are a new technology that has recently begun to be put to practical use, and the diagnostic equipment that can be equipped with these probes is limited to the high-performance models of each manufacturer.
On the other hand, annular array probes can be designed specifically for each observation task, and ultrasound transmission, reception, and echo data accumulation can be performed by a simple system. Also, unlike commercial diagnostic equipment for clinical use, the accumulated echo data is a complete raw signal without any filtering and is significant in signal analysis. In other words, the ultrasound system including an annular array probe as proposed in this study has a simple structure, and its manufacturing cost is extremely low (even for prototypes that require a large development budget, the price is at least 1/10th of that of ordinary clinical devices) compared to advanced ultrasound diagnostic systems. The system can be easily introduced to clinical departments that are not familiar with ultrasound diagnosis. The ease of operation and user-friendly interface (optimized under the supervision of plastic surgeons and dermatologists) of our simple ultrasound system with a specially designed annular array probe would also be effective in expanding the range of clinical applications.
The above explanation was described in Lines *64-82* in the main text.
Comments 2. Enhanced Statistical Analysis of BSC Data
- The current analysis of the relationship between BSC deviations and scatterer diameters can be further refined. Applying more sophisticated statistical models or machine learning techniques could improve the predictive accuracy of the BSC evaluations and enhance the robustness of the study’s findings. This would allow the manuscript to offer deeper insights into the data and present more reliable conclusions.
Response2.
The problem for clinical application is that there are multiple types of scatterers in actual biological tissues, and the Faran model used for BSC evaluation in this study may deviate from the actual scatterer structure. However, this problem is not limited to annular array probes, but occurs in any ultrasonic signal analysis using any sensor, so a comprehensive study on the matching of theory (numerical models) and measurement (actual echo analysis) should be continued. The challenges limited to annular array probes are the limited length of the DOI due to the small number of sensors and the difficulty of ensuring observation accuracy in deep areas. It is assumed that these problems can be addressed by varying the maximum diameter of the sensor and the thickness of each ring depending on the depth and size of the area to be observed. We are currently designing a 5-channel annular array probe with a maximum diameter of 30 mm for verification, and will promote further studies in phantoms and real skin tissues in the future.
The above explanation has been added at the end of Conclusion.
In fact, we, as well as other BSC researchers, have already compared the accuracy of BSC evaluations using multiple mathematical models and proposed new mathematical models. However, since the purpose of this study was to investigate the basic potential stresses for BSC evaluation in annular array probes, the Faran model, which is the most common and frequently used in clinical applications, was used. This approach is the most common way to validate new technologies (hardware and algorithms) in the area of BSC evaluation.
Machine learning-based studies are becoming the standard for quantitative diagnosis using a variety of medical imaging modalities and biometric methods. However, brushing up measurement and signal analysis techniques and considering AI diagnosis at the same time should be avoided. This is because it is extremely difficult to determine which effects have improved diagnostic accuracy. Therefore, again, according to the general logic of this research area, machine learning considerations are not included in this study. The study will be conducted after a large amount of data can be accumulated using annular array probes.
In fact, we have proposed a novel evaluation method for QUS of the liver, promoted its clinical application, and subsequently pursued the possibility of AI diagnosis.
Nguyuen, T. N.; Podkowa, A. S.; Park, T.H.; Miller, R. J.; Do, M. N.; Oelze, M. L.; Use of a convolutional neural network and quantitative ultrasound for diagnosis of fatty liver. Ultrasound Med Biol. 2021, Vol. 47(3), pp. 556-568
Comments 3. Discussion of Potential Methodological Limitations
- A more detailed discussion of the potential limitations within the experimental setup or data interpretation would greatly benefit the manuscript. Addressing issues such as assumptions made during the BSC calculations, the consistency of phantom materials, or the impact of specific probe characteristics on the results would provide a more balanced view of the study’s reliability. This critical evaluation would help readers understand the context and scope of the findings better.
Response3.
As mentioned above, we added it to Conclusion.
Comments 4. Addressing High-Frequency Attenuation Issue
- The manuscript could be strengthened by discussing possible alternative approaches to mitigate high-frequency attenuation challenges, particularly in phantoms with larger scatterer diameters. Exploring different probe designs, materials, or advanced signal processing techniques could provide practical solutions to these challenges, making the study’s conclusions more robust and applicable to various scenarios.
Response4.
As mentioned above, We have added a note in the Conclusion that future studies using probes different from those used in this study are needed.
As you commented, it is obvious that changing the probe also changes the performance of BSC evaluation. However, since this study uses a probe for which we have already confirmed excellent imaging performance, it is possible to objectively understand the relationship between the ultrasonic sound field and BSC evaluation results.
Based on the results of this study, annular array probes have been designed with different materials, maximum diameters, etc. as you have pointed out for future studies. Specifically, composite material is used as the element to allow strong ultrasonic waves to enter deep areas, and the maximum diameter is 30 mm (this part of the explanation cannot be published in the paper). Creating such a new annular array probe is such a labor-intensive task that it alone is a research challenge. It takes a minimum of six months to actually complete this probe and use it for BSC evaluation. It also requires a large budget because it is a completely new prototype probe. Therefore, we mention the possibility of annular array probes in this paper, and would like to disclose the development including new probes as a separate study. This is also the usual research logic.
Comments 5. Exploration of Alternative Probe Configurations
- Consider including a discussion on the potential of alternative configurations for the annular-array probe. The manuscript could explore the theoretical impact of varying the number of elements in the array, different element arrangements, or different materials used in the probe’s construction. Such an exploration could broaden the scope of the study, suggesting new directions for future research and development.
Response5.
The response to comment 5 is summarized in the response to comment 4.
Comments 6. Development of a More User-Friendly Interface for Probe Operation
- The manuscript would benefit from discussing the development of a more intuitive and user-friendly interface for the annular-array probe. This could involve integrating real-time data processing capabilities, providing immediate feedback to the operator, or simplifying the interface to enhance usability in clinical settings. Addressing this aspect could increase the practical relevance of the study and encourage broader adoption of the technology.
Response6.
This point has been added in conjunction with the response to comment 1.
These suggestions aim to enhance the manuscript by deepening the analysis, addressing potential limitations, and proposing practical improvements. Incorporating these elements would provide a more comprehensive and impactful contribution to the field.
In the following, it is not mandatory.
If you are considering the possibility of conducting additional experiments to further enhance the study, it would be valuable to explore a few key areas. Below are suggestions for potential experiments that could provide deeper insights and strengthen the findings:
Comments7. Broader Range of Tissue-Mimicking Phantoms
- Objective: To validate the generalizability of the annular-array probe across a wider variety of tissue types.
- Proposed Experiment: Create and evaluate additional phantoms that mimic a broader spectrum of biological tissues, including more complex and heterogeneous structures. This could include phantoms with varying degrees of scatterer density, size, and distribution to better simulate real tissue conditions and assess the probe’s performance across different scenarios.
Response7.
The points raised are extremely important and interesting to us, and we would of course like to put these considerations into practice. In fact, we have also evaluated heterogeneous media as a basic study, but it is extremely difficult to judge the validity of such evaluations, no matter what the medical imaging modality. Therefore, we would like to give priority to the publication and discussion of the evaluation results in homogeneous media first.
In other words, we plan to systematically practice “probe change,” “model change,” and “evaluation target change” in the next steps.
Comments8. Longitudinal Stability of BSC Measurements
- Objective: To evaluate the stability and repeatability of BSC measurements over time.
- Proposed Experiment: Conduct a longitudinal study where the BSC of a specific phantom is measured at multiple time points. This would help determine the consistency and reliability of the BSC values over time, which is crucial for monitoring disease progression or the effects of treatments.
Response8.
The temporal variation of the observed object and the reproducibility of the measurements are also interesting topics, as we practice them in our shear wave elastography studies and others *. For standardization studies, it is basically important to develop a phantom that does not undergo temporal degeneration, but it is also meaningful to use a subject that changes, and this will be added to future studies. (Studies in animal models with different pathological conditions have already been scheduled.)
* Kishimoto, R.; Suga, M.; Usumura, M.; Iijima, H.; Yoshida, M.; Hachiya, H.; Shiina, T.; Yamakawa, M.; Konno, K.; Obata, T.; Yamaguchi, T.; Shear wave speed measurement bias in a viscoelastic phantom across six ultrasound elastography systems: a comparative study with transient elastography and magnetic resonance elastography. Journal of Medical Ultrasonics 2022, Vol.49, pp. 143-152
Comments9. Comparison with In Vivo Measurements
- Objective: To bridge the gap between in vitro phantom studies and real-world clinical applications.
- Proposed Experiment: If feasible, perform preliminary in vivo studies using animal models or human subjects to compare the BSC measurements obtained from phantoms with those obtained from actual biological tissues. This would provide a clearer understanding of how well the annular-array probe performs in a clinical setting and whether the results from phantom studies can be reliably extrapolated to human tissues.
Response9.
As mentioned above, animal models are being prepared and applications for clinical studies have been submitted. However, as is well understood, there are many constraints and time-consuming processes to initiate these practices. It is our intention to conduct these studies in parallel with the brushing up of the measurement system. In particular, clinical studies should be conducted after optimization of the transmitter/receiver system is completed.
Comments10. High-Frequency Attenuation Mitigation
- Objective: To explore solutions for the high-frequency attenuation issues observed in certain phantoms.
- Proposed Experiment: Experiment with different probe materials, configurations, or signal processing techniques to see if these can reduce attenuation effects at higher frequencies. This could involve testing alternative membrane materials or adjusting the geometry of the annular-array probe to optimize performance in challenging conditions.
Response10.
As mentioned above, a new probe is being designed and prototype production has begun. This study is also exciting for us and we hope to complete it as soon as possible. Of course, it is necessary to evaluate the sensor characteristics of each channel and confirm the beam characteristics in advance of the actual measurement study.
Comments11. Alternative Probe Configurations
- Objective: To determine the optimal configuration of the annular-array probe for various clinical applications.
- Proposed Experiment: Test different configurations of the annular-array probe, such as varying the number of elements, the arrangement of those elements, or the curvature of the array. By systematically varying these parameters, it would be possible to identify configurations that offer the best balance of resolution, depth of field, and sensitivity for specific types of tissue.
Response11.
Same as the response10. Since designing just one new probe requires time and budgetary costs, we assume that one to three prototype probes and computer simulations will be used to address this issue.
Comments12. Evaluation of User Interface Usability
- Objective: To assess and improve the usability of the probe’s interface in clinical settings.
- Proposed Experiment: Conduct a usability study with clinicians to evaluate the current interface of the annular-array probe. Collect feedback on ease of use, real-time data interpretation, and workflow integration. Based on this feedback, develop and test improvements that could enhance the user experience and potentially increase the adoption of the technology in clinical practice.
Response12.
We think this is a very important consideration for practical use. We are collaborating with plastic surgeons, dermatologists, ophthalmologists, gastroenterologists, and gastroenterologists. Since plastic surgery and dermatology are in particular need of the technology proposed in this study, we would like to study the optimization of the user interface under the supervision of several physicians in those fields. However, we will not take the lead in this study, but rather, in cooperation with experts in human interface and computer vision, we will ask them to promote it separately as a new study. Their studies will be of great benefit in their research area.
If any of these proposed experiments are feasible, they could significantly enhance the robustness of the study and provide more comprehensive data to support the conclusions. Please let me know if you would like to explore these options further or if there are specific aspects you are interested in investigating.
Thank you for proposing an extremely grand and comprehensive research plan. We will ensure the completion of the basic study and promote new studies one after another.
Round 2
Reviewer 2 Report
Comments and Suggestions for Authors
Thanks to authors for their implementation the manuscript
Author Response
Comments: Thanks to authors for their implementation the manuscript
Reply: Thank you for reviewing and accepting our manuscript.
Reviewer 3 Report
Comments and Suggestions for Authors
Accept
Author Response
Comments: Accept
Reply: Thank you for reviewing and accepting our manuscript.